# The BDNF Val66Met Polymorphism (rs6265) Modulates Inflammation and Neurodegeneration in the Early Phases of Multiple Sclerosis

**DOI:** 10.3390/genes13020332

**Published:** 2022-02-10

**Authors:** Ettore Dolcetti, Antonio Bruno, Federica Azzolini, Luana Gilio, Alessandro Moscatelli, Francesca De Vito, Luigi Pavone, Ennio Iezzi, Stefano Gambardella, Emiliano Giardina, Rosangela Ferese, Fabio Buttari, Francesca Romana Rizzo, Roberto Furlan, Annamaria Finardi, Alessandra Musella, Georgia Mandolesi, Livia Guadalupi, Diego Centonze, Mario Stampanoni Bassi

**Affiliations:** 1Neurology Unit, IRCSS Neuromed, 86077 Pozzilli, Italy; dolcettiettore@gmail.com (E.D.); brunoa.neuro@gmail.com (A.B.); federica.azzolini@gmail.com (F.A.); gilio.luana@gmail.com (L.G.); f.devito.molbio@gmail.com (F.D.V.); gi19gi82@gmail.com (L.P.); ennio.iezzi@neuromed.it (E.I.); stefano.gambardella@uniurb.it (S.G.); ferese.rosangela@gmail.com (R.F.); fabio.buttari@gmail.com (F.B.); rizzo.francescaromana@gmail.com (F.R.R.); m.stampanonibassi@neuromed.it (M.S.B.); 2Department of Systems Medicine, Tor Vergata University, 00133 Rome, Italy; a.moscatelli@hsantalucia.it (A.M.); livia.guadalupi@gmail.com (L.G.); 3Laboratory of Neuromotor Physiology, IRCSS Fondazione Santa Lucia, 00179 Rome, Italy; 4Department of Biomolecular Sciences, University of Urbino “Carlo Bo”, 61029 Urbino, Italy; 5Genomic Medicine Laboratory, IRCCS Fondazione Santa Lucia, 00179 Rome, Italy; emiliano.giardina@uniroma2.it; 6Department of Biomedicine and Prevention, University of Rome “Tor Vergata”, 00133 Rome, Italy; 7Clinical Neuroimmunology Unit, Institute of Experimental Neurology (INSpe), Division of Neuroscience, San Raffaele Scientific Institute, 20132 Milan, Italy; furlan.roberto@hsr.it (R.F.); finardi.annamaria@hsr.it (A.F.); 8Synaptic Immunopathology Lab, IRCCS San Raffaele Roma, 00163 Rome, Italy; alessandra.musella@uniroma5.it (A.M.); georgia.mandolesi@uniroma5.it (G.M.); 9Department of Human Sciences and Quality of Life Promotion, University of Rome San Raffaele, 00163 Rome, Italy

**Keywords:** BDNF, Val66Met, inflammation, neurodegeneration, multiple sclerosis, rs6265, cytokines

## Abstract

The clinical course of multiple sclerosis (MS) is critically influenced by the interplay between inflammatory and neurodegenerative processes. The brain-derived neurotrophic factor (BDNF) Val66Met polymorphism (rs6265), one of the most studied single-nucleotide polymorphisms (SNPs), influences brain functioning and neurodegenerative processes in healthy individuals and in several neuropsychiatric diseases. However, the role of this polymorphism in MS is still controversial. In 218 relapsing–remitting (RR)-MS patients, we explored, at the time of diagnosis, the associations between the Val66Met polymorphism, clinical characteristics, and the cerebrospinal fluid (CSF) levels of a large set of pro-inflammatory and anti-inflammatory molecules. In addition, associations between Val66Met and structural MRI measures were assessed. We identified an association between the presence of Met and a combination of cytokines, identified by principal component analysis (PCA), including the pro-inflammatory molecules MCP-1, IL-8, TNF, Eotaxin, and MIP-1b. No significant associations emerged with clinical characteristics. Analysis of MRI measures evidenced reduced cortical thickness at the time of diagnosis in patients with Val66Met. We report for the first time an association between the Val66Met polymorphism and central inflammation in MS patients at the time of diagnosis. The role of this polymorphism in both inflammatory and neurodegenerative processes may explain its complex influence on the MS course.

## 1. Introduction

Multiple sclerosis (MS) is a chronic inflammatory disease of the central nervous system (CNS) that is nowadays representing one of the main causes of disability in young people [1]. The clinical course of MS is highly variable and reflects the complex pathogenesis characterized by inflammation and neurodegeneration already detectable in the initial stages of the disease. The interplay between these two processes is critically regulated by the expression of specific inflammatory mediators and neurotrophins [2]. Pro-inflammatory cytokines and chemokines released by autoreactive lymphocytes and activated resident immune cells drive neuroinflammation and the formation of demyelinating white matter lesions. Gray matter involvement in MS is not merely a consequence of chronic axonal damage; it has been proposed that inflammation may play a causal role, a process defined as inflammatory neurodegeneration [2,3]. Accordingly, studies in animal models of MS (i.e., experimental autoimmune encephalomyelitis, EAE) have shown that pro-inflammatory molecules may alter synaptic transmission-promoting neuronal hyperexcitability and glutamatergic excitotoxicity [3,4]. Furthermore, elevated cerebrospinal fluid (CSF) levels of specific pro-inflammatory cytokines have been associated with worse long-term disability and increased neurodegeneration in patients with MS [5,6].

It is conceivable that individual genetic variability influencing the processes of inflammation and neurodegeneration may contribute to clinical heterogeneity in MS.

The brain-derived neurotrophic factor (BDNF) gene is located on chromosome 11p13, and the variant rs6265 (NM_001143810.1:c (196G > A) on ref genomes GRCh38/hg38, Human Genome Variant Society, http://www.hgvs.org/mutnomen) represents one of the most widely studied single-nucleotide polymorphisms (SNPs). This SNP, known as “Val66Met”, produces a valine (Val) to methionine (Met) change at position 66 (p.Val66Met) of the proBDNF protein (NP_001137282.1). Although it is classified as a benign polymorphism according to ACMG guidelines [7], several reports consider rs6265 a possible risk factor, as related to the impairment in BDNF activity-dependent BDNF secretion [8]. This polymorphism has been involved in numerous neuropsychiatric diseases, including dementia, schizophrenia, and anxiety–depressive spectrum disorders [8,9,10]. In MS, the role of the BDNF Val66Met polymorphism is still controversial [11,12]. The close relationship between inflammation and neurodegeneration in MS may contribute to these contrasting reports. Notably, in addition to its antiapoptotic and neuroprotective activity, it has recently been reported that BDNF may exert anti-inflammatory effects [13].

To explore whether the BDNF Val66Met polymorphism could influence the inflammatory response in MS, we analyzed the associations between this SNP and the CSF levels of a large set of pro-inflammatory and anti-inflammatory molecules in a group of relapsing–remitting (RR)-MS patients at the time of diagnosis.

## 2. Materials and Methods

### 2.1. MS Patients

A group of 218 consecutive RR-MS Italian patients from Central and Southern Italy were enrolled in the study. Patients were admitted to the neurological clinic of the Neuromed Research Institute in Pozzilli, Italy, between 2016 and 2019 and diagnosed with MS on the basis of clinical, laboratory, and MRI parameters [14]. The Ethics Committee of Neuromed Research Institute approved the study (cod. 06-17) according to the Declaration of Helsinki. All patients provided written informed consent to participate in the study. At the time of diagnosis, patients underwent clinical evaluation and brain and spine MRI. Clinical characteristics included age, sex, expanded disability status score (EDSS), the presence of radiological disease activity, and disease duration, which was measured as the interval between disease onset and diagnosis.

### 2.2. SNP Val66Met Analysis

Genotyping for BDNF SNP Val66Met was performed in all enrolled patients. A blood sample (200 μL) was collected at the time of diagnosis. Genomic DNA was isolated from peripheral blood leukocytes according to standard procedures (QIAamp DNA Blood Mini Kit, QIAGEN LLC, Germantown, MD, USA). The BDNF region containing the Val66Met polymorphism was analyzed with a TaqMan Validate SNP Genotyping Assay (Applied Biosystems, Foster City, CA, USA) using the ABI-Prism 7900HT Sequence Detection System (Applied Biosystems, Foster City, CA, USA) from 25 ng of genomic DNA in a final volume of 15 μL, according to the manufacturer’s instructions.

### 2.3. CSF Collection and Analysis

In 210 MS patients, the CSF concentrations of inflammatory cytokines were analyzed. CSF was collected at the time of diagnosis, during hospitalization, by lumbar puncture (LP). No corticosteroids or disease-modifying therapies were administered before LP. The CSF samples were stored at −80 °C and later analyzed using a Bio-Plex multiplex cytokine assay (Bio-Rad Laboratories, Hercules, CA, USA). CSF cytokines levels were determined according to a standard curve generated for the specific target and expressed as picograms/milliliter (pg/mL). Samples were analyzed in triplicate. The CSF cytokines analyzed included interleukin (IL)-1β, IL-2, IL-4, IL-5, IL-6, IL-7, IL-8, IL-9, IL-10, IL-12, IL-13, IL-15, IL-17, tumor necrosis factor-α (TNF), interferon-γ (IFN-**γ**), macrophage inflammatory protein (MIP-1a, MIP-1b), monocyte chemoattractant protein (MCP-1), granulocyte colony stimulating factor (G-CSF), granulomonocyte colony stimulating factor (GM-CSF), interleukin-1 receptor antagonist (IL-1ra), Eotaxin, fibroblast growth factor (FGF), interferon γ-induced protein 10 (IP-10), platelet-derived growth factor (PDGF), regulated upon activation, normal T cell expressed and secreted (RANTES), and vascular endothelial growth factor (VEGF).

### 2.4. MRI

All the patients underwent a 1.5T MRI scan, which included the following sequences: dual-echo proton density, fluid-attenuated inversion recovery (FLAIR), T1-weighted spin-echo (SE), T2-weighted fast SE, and contrast-enhanced T1-weighted SE before and after intravenous gadolinium (Gd) infusion (0.2 mL/kg). Radiological disease activity at the time of diagnosis was defined as the presence of Gd-enhancing (Gd+) lesions at the time of hospitalization. All the acquired sequences were performed by General Electric Signa HDXT MRI equipped with an 8-channel head coil.

In 68 patients, an additional 3T MRI scan was performed to explore the association between the BDNF Val66Met polymorphism and structural MRI measures. The MRI scan included a 3D Spoiled Gradient Recalled (SPGR) T1-weighted sequence (178 contiguous sagittal slices, voxel size 1 × 1 × 1 mm, TR 7 ms, TE 2.856 ms, inversion time 450 ms) and a 3D FLAIR CUBE sequence (208 contiguous sagittal 1.6 mm slices, voxel size, 0.8 × 0.8 × 0.8 mm, TR 6000 ms, TE 139.45 ms, inversion time 1827 ms). All the sequences acquired were performed by General Electric Signa HDXT Twin Speed MRI equipped with an 8-channel head coil.

Cortical thickness (CT) was computed from the lesion-filled 3D T1 image by using the computational anatomy toolbox (CAT12, version 916, https://dbm.neuro.uni-jena.de/cat/), as implemented in SPM12 (https://www.fl.ion.ucl.ac.uk/spm). Lesion filling was performed by segmenting the white matter lesions from FLAIR and T1 images by using the lesion growth algorithm as implemented in the lesion segmentation tool (www.statistical-modelling.de/lst.html, version 2.0.15) for SPM12. Finally, T2 lesion load was computed from 3D T1 and 3d FLAIR images by using a well-established pipeline [15].

### 2.5. Statistical Analysis

The Shapiro–Wilk test was used to evaluate the normality distribution of continuous variables. The data were shown as mean (standard deviation, SD) or median (interquartile range, IQR). Categorical variables were presented as absolute (n) and relative frequency (%). Chi-square or, when necessary, Fisher exact test was employed to explore the association between categorical variables. The difference in continuous variables between the BDNF SNP groups was evaluated using the nonparametric Mann–Whitney test. A *p* value ≤ 0.05 was considered statistically significant. Box plots were employed to highlight statistically significant differences between groups. All the comparisons were performed using IBM SPSS Statistics for Windows/Mac (IBM Corp., Armonk, NY, USA). Figures were made with Prism GraphPad 6.0.

Spearman’s nonparametric correlation and partial correlation were used to test the possible associations between variables that were not normally distributed. Principal Component Analysis (PCA) was applied to the samples of the 27 CSF cytokines. PCA allowed us to reduce the dimensionality of the data set and to study possible synergic effects of the individual cytokines. The first nine PCs explained more than 80% of the variance and were retained for further analysis (see below). Next, we used a logistic regression to test the association between the proportion of the Val/Val on the BDNF Val66Met polymorphism and the score on the first nine PCs. Finally, we used a logistic regression to test the association between the proportion of the Val/Val and individual cytokines selected from the previous analysis, correcting for the effects of disease duration.

## 3. Results

### 3.1. Role of BDNF Gene in Disease Characteristics of MS Patients

The demographic and clinical characteristics of patients are shown in Table 1.

Allele and genotype frequencies of rs6265 of our MS cohort were in Hardy–Weinberg equilibrium when considering the data from the Gnomad database regarding European population (frequency of non-risk allele G = 0.811; frequency of risk allele A = 0.189, chi-square n.s. *p* = 0.199).

In our MS cohort, the genotypes frequencies were represented by AA (*n* = 12; 5.50%, Met/Met patients), AG (*n* = 70; 32.11%, Val/Met patients), and GG (*n* = 136; 62.38%; Val/Val patients). To obtain two comparable groups, we unified the Met/Met and Val/Met patients in a single group (Met carriers) (Table 2).

No significant differences emerged between the two groups in clinical characteristics, specifically for age at diagnosis (*p* = 0.269), sex distribution (*p* = 0.052), EDSS at diagnosis (*p* = 0.390), the presence of radiological activity (*p* = 0.078), and presence of OCB (*p* = 0.795). We only found a small difference between the two groups in disease duration (Met carriers, median (IQR): = 0.4 (0.1–2) vs. Val/Val, median (IQR) = 0.2 (0.1–1.1), *p* = 0.039).

### 3.2. Role of BDNF Gene in the Regulation of Pro-Inflammatory Cytokine Levels in MS Patients

To explore whether individual genetic variability in the BDNF gene could influence central inflammation in MS, we analyzed the possible association between Val66Met and the CSF cytokines’ profile.

We ran principal component analysis on 27 cytokines The first nine principal components explained 80% of the variance, suggesting a synergistic effect of the different cytokines (Appendix A, see below). The association of specific cytokines with the first four PCs are shown in Figure 1a,b.

We used logistic regression to test for the association between the Val/Val on the polymorphism of the BDNF gene and the first nine PCs. We found a significant association with PC3 (*p* = 0.017) and a non-significant trend with PC4 (*p* = 0.05). As shown in Figure 1b, the following cytokines had high negative loading on PC3: MIP-1b, Eotaxin, TNF, IL-8, and MCP-1, among others.

Comparing cytokines that belong to the PC3 group, in Met carriers, we found higher CSF levels of TNF (Met carriers, median (IQR): = 2.8 (1.5–4.3) vs. Val/Val, median (IQR) = 1.95 (0.8–3.4), *p* = 0.019), IL-8 (Met carriers, median (IQR): = 23.8 (16.4–30.6) vs. Val/Val, median (IQR) = 19.4 (14.1–27), *p* = 0.006), and MCP-1 (Met carriers, median (IQR): = 139.4 (102.4–191.55) vs. Val/Val, median (IQR) = 120.8 (91.6–160.3), *p* = 0.011). Conversely, no significant differencies were found in CSF levels of Eotaxin (Met carriers, median (IQR): = 0.9 (0.6–1.3) vs. Val/Val, median (IQR) = 0.8 (0.5–1.2), *p* = 0.173), and MIP-1b (Met carriers, median (IQR): = 4.3 (3.2–6.4) vs. Val/Val, median (IQR) = 4.5 (3.3–7), *p* = 0.727) between the two groups (Figure 2).

Finally, we used logistic regression to test the association between individual cytokines with high loading in PC3 (cut-off: absolute value of loading > 2) and the proportion of Val/Val. In order to exclude an effect of disease duration, we included this variable in the logistic regression as a predictor. We found a significant association with MCP-1 (*p* = 0.006); the probability of observing Val/Val was smaller for a higher concentration of MCP-1.

### 3.3. Role of BDNF Gene on MRI Structural Measurements in MS Patients

We investigated the association between Val66Met and MRI structural measures at the time of MS diagnosis. We found reduced cortical thickness (CT) in Met carriers (Met carriers, median (IQR): = 2.6 (2.5–2.7) vs. Val/Val, median (IQR) = 2.7 (2.6–2.8), *p* = 0.042). Conversely, no differences were found in T2-weighted lesion load (Met carriers, median (IQR): = 0.5 (0.2–1.9) vs. Val/Val, median (IQR) = 0.3 (0.1–1.7), *p* = 0.400) (Figure 3).

## 4. Discussion

The Val66Met polymorphism, characterized by a valine-to-methionine substitution at codon 66, is associated with altered intracellular trafficking and secretion of BDNF in the synaptic cleft [8,16] in both homozygotes and heterozygotes individuals [17]. BDNF is a major member of the neurotrophin family, involved in neuronal growth and differentiation. Accordingly, BDNF released in response to neuronal activity critically regulates glutamatergic transmission and synaptic plasticity [18,19]. BDNF is also actively secreted by the immune cells of RR-MS patients during relapses and in the recovery phases [20] and may have a role in limiting the negative effects of neuroinflammation and promoting compensation from brain damage [21,22].

In the present study, we provided the first evidence of an association between the Val66Met polymorphism and neuroinflammation in the early stages of MS. The allele coding for Met was significantly associated with a linear combination of cytokines (principal component) identified by PCA, such as MCP-1, IL-8, TNF, Eotaxin, and MIP-1b. Inflammatory mediators play an important role in the induction of the brain’s inflammatory response in MS. MCP-1, IL-8, TNF, Eotaxin, and MIP-1b are involved in the pathogenesis of MS, and elevated CSF levels of some of these molecules, particularly IL-8 and TNF, have previously been associated with disease reactivation and progression [4,6,23] and have been directly implicated in inflammatory neurodegeneration [6]. MCP-1 showed the strongest association with Val66Met in our study. This chemokine is secreted by several immune cells and promotes the chemotaxis of macrophage/microglial cells [24,25]. MCP-1 is expressed in acute and chronic demyelinating lesions [26] and may play a role in MS progression [27].

Our results suggest that MS patients carrying the allele coding for Met may present with higher levels of central inflammation at diagnosis. Experimental studies have demonstrated that BDNF may exert anti-inflammatory effects [13]. It has been reported that the administration of BDNF reduced clinical disability in EAE, attenuating neuronal death, and central inflammation [28]. In addition, in a rat model of ischemic–hypoxic damage, BDNF administration reduced inflammation and, particularly, the levels of TNF [29]. In an experimental model of inflammation, absent BDNF expression in the cerebral cortex and hippocampus has been associated with neuronal damage, microglial/macrophage proliferation, and elevated levels of the pro-inflammatory molecules TNF, IL-1β, and IL-6 [30]. Furthermore, BDNF administration decreased the mRNA and protein levels of these cytokines and increased the expression of anti-inflammatory molecules [13]. Ischemic damage in mice carrying the human BDNF Val66Met polymorphism triggered inflammatory responses, with the prevalence of pro-inflammatory macrophages showing enhanced activation and migration. Importantly, similar characteristics were observed in macrophages from Met/Met patients with coronary heart disease [31]. Taken together, these studies indicate that BDNF may exert direct anti-inflammatory activities and that Val66Met may be associated with a pro-inflammatory immune response. In line with these findings, our results suggest that in MS patients carrying the allele coding for Met, altered BDNF expression may be associated with a pro-inflammatory CSF milieu at the time of diagnosis.

In the present study, no significant associations emerged between the Val66Met polymorphism and clinical features of MS at diagnosis. The influence of this polymorphism on MS characteristics and clinical course is still controversial [32,33,34]. Although Val66Met has been associated with MS susceptibility and disability [35], beneficial effects have also been reported [36,37]. A variability in outcome measures as well as in clinical phenotype and patient characteristics could partially explain discrepancies between studies. Different levels of central inflammation in these patients might represent an additional source of variability.

The analysis of structural MRI measures showed reduced cortical thickness in Val66Met MS patients at diagnosis. Previous studies investigating the role of this polymorphism on MRI structural measures in MS have yielded contrasting results. A previous work in RR-MS evidenced reduced gray matter volumes in Met carriers [38]. However, other studies reported a protective effect of Val66Met on cortical and hippocampal volumes [11,39,40,41]. This variability contrasts with the data obtained in healthy subjects. In this context, Val66Met has been linked to decreased cortical volumes in specific brain areas, including the hippocampus and the prefrontal and cingulate cortex [42,43,44,45]. To explain these paradoxical findings, it has been proposed that excessive BDNF expression may lead to a chronic enhancement of NMDA-dependent glutamatergic synaptic transmission [18,46,47,48], further promoting excitotoxic neurodegeneration in MS [40]. Notably, in MS, de-methylation of the BDNF gene has been observed in response to increased inflammatory activity [49]. In MS patients, Val66Met may therefore have a protective function against the possible negative effects of excessive BDNF expression [40].

In our study, the association between Val66Met and lower cortical thickness suggests that in the early stages of MS the effect of this polymorphism is similar to that observed in healthy subjects. The protective effects of Val66Met against excitotoxic neurodegeneration may become apparent during the course of the disease. Although the low number of patients with structural MRI measures does not allow us to formulate firm conclusions and does not permit investigating other parameters such as volumes of specific brain areas, our results were obtained in a cohort of newly diagnosed MS patients with homogeneous clinical phenotype, low disability, and short disease duration. The lack of prospective MRI and clinical data is a major limitation of the present study and further research is needed to assess the impact of the Val66Met polymorphism during the course of MS.

In conclusion, we report for the first time an association between the BDNF Val66Met polymorphism and central inflammation in MS patients at the time of diagnosis. Our findings suggest a role for this polymorphism in both inflammatory and neurodegenerative processes and may contribute to explaining its complex influence on the MS course.

## Figures and Tables

**Figure 1 genes-13-00332-f001:**
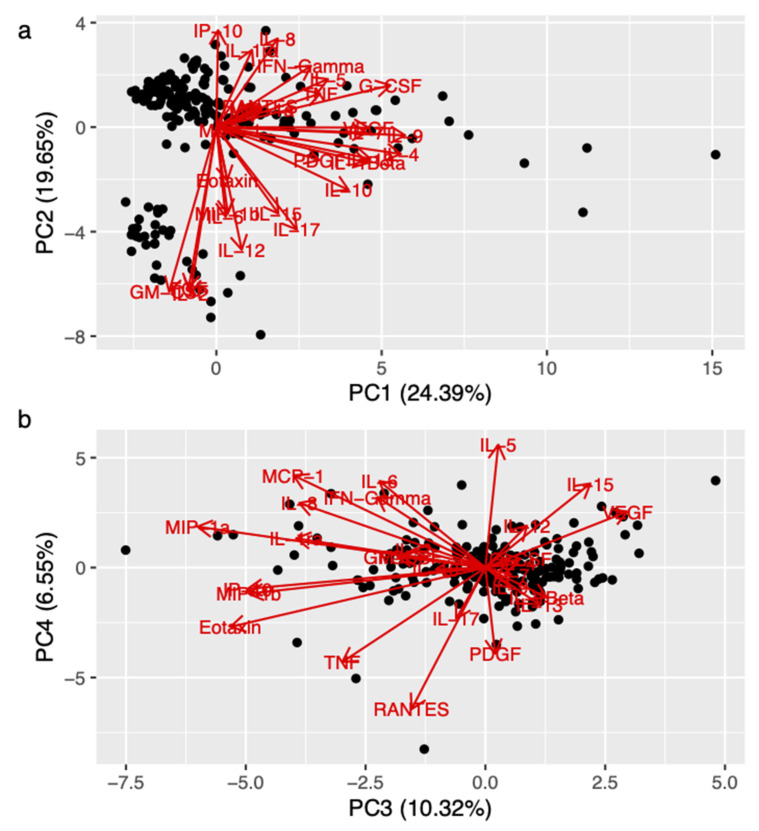
Association between synergic effect of cytokines (principal component, PC) and BDNF Val66Met. Figure 1 (biplot) shows the orientation of the different cytokines with respect to the first and the second component (panel (**a**)), and with respect to the third and the last component (panel (**b**)), respectively. The biplots showing the orientation of the cytokines with respect to the first four PCs. Legend: FGF (fibroblast growing factor); G-CSF (granulocyte colony stimulating factor); GM-CSF (granulomonocyte colony stimulating factor); IFN-γ (interferon-γ); IL (interleukin); IL-ra (interleukin-1 receptor antagonist); IP-10 (interferon γ-induced protein 10); MCP (monocyte chemoattractant protein); MIP (macrophage inflammatory protein); PC (principal component); PDGF (platelet-derived growth factor); RANTES (regulated upon activation, normal T cell expressed and secreted); TNF (tumor necrosis factor); VEGF (vascular endothelial growth factor).

**Figure 2 genes-13-00332-f002:**
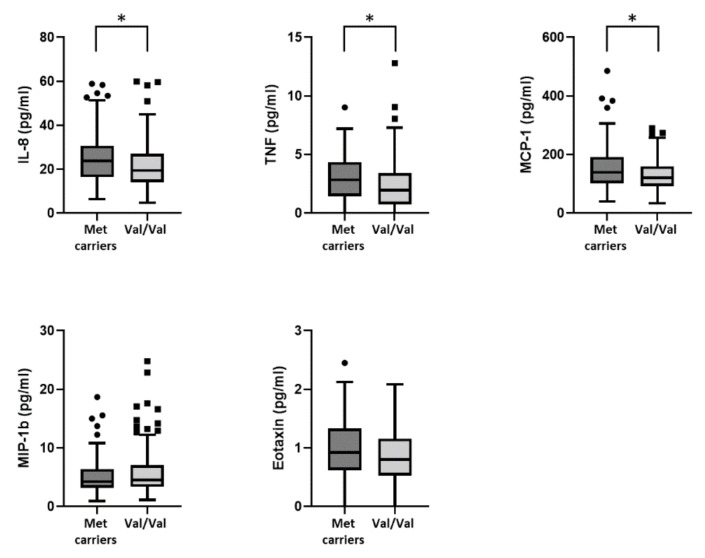
Association between BDNF Val66Met polymorphism and the CSF levels of TNF, IL-8, MCP-1, Eotaxin, and MIP-1b. Figure 2 legend: CSF (cerebrospinal fluid); IL (interleukin); MCP-1 (monocyte chemoattractant protein-1); MIP-1b (macrophage inflammatory protein 1b); TNF (tumor necrosis factor). (*) denotes statistical significance (*p* < 0.05).

**Figure 3 genes-13-00332-f003:**
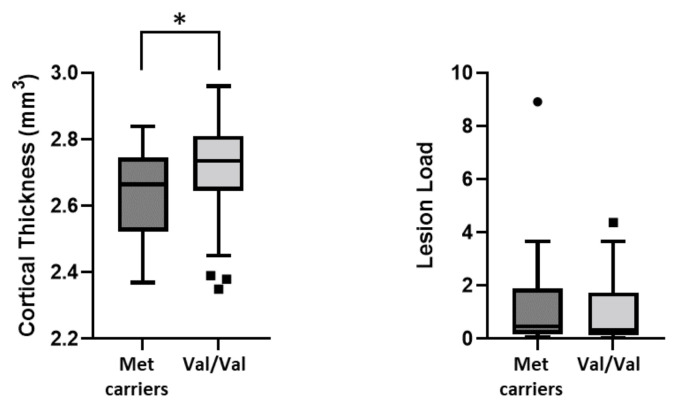
Association between BDNF Val66Met polymorphism and structural MRI measures. Legend: (*) denotes statistical significance (*p* < 0.05).

**Table 1 genes-13-00332-t001:** Clinical characteristics of MS patients.

RR-MS	N	218
Age at diagnosis, years	Median (IQR)	34 (25.9–44.8)
Sex, F/M	N (%)	155/63 (71.1/28.9)
Disease duration, years	Median (IQR)	0.25 (0.06–1.4)
EDSS at diagnosis	Median (IQR)	1.5 (1–2)
Radiological activity, yes	N/tot (%)	87/206 (42.2)
OCB, yes	N/tot (%)	165/214 (77.1)

Table 1 Legend: expanded disability status scale (EDSS); female (F); interquartile range (IQR); male (M); multiple sclerosis (MS); number (N); oligoclonal bands (OCB); Relapsing–remitting (RR).

**Table 2 genes-13-00332-t002:** Clinical characteristics of MS patients according to BDNF Val66Met polymorphism.

RR-MS	N	Met Carriers (82)	Val/Val (136)	*p*
Age at diagnosis, years	Median (IQR)	36 (26.8–45.3)	33.3 (25.2–44.1)	0.269
Sex, F/M	N (%)	52/30 (63/37)	103/33 (78/24)	0.052
Disease duration, years	Median (IQR)	0.4 (0.1–2)	0.2 (0.1–1.1)	0.039 ***
EDSS at diagnosis	Median (IQR)	1.5 (1–2)	1.5 (1–2)	0.390
Radiological activity, yes	N/tot (%)	39/78 (50)	48/128 (37.5)	0.078
OCB, yes	N/tot (%)	64/82 (78)	101/132 (76.5)	0.795

Table 1 Legend: expanded disability status scale (EDSS); female (F); interquartile range (IQR); male (M); multiple sclerosis (MS); number (N); oligoclonal bands (OCB); Relapsing–remitting (RR). (*) denotes statistical significance (*p* < 0.05) using a nonparametric Mann–Whitney test for continuous variables and Chi-square for categorial variables.

## Data Availability

Anonymized datasets are available upon reasonable request to the corresponding author.

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
