# Peer review of "The BDNF Val66Met Polymorphism (rs6265) Modulates Inflammation and Neurodegeneration in the Early Phases of Multiple Sclerosis"

_genes, 2022, doi:10.3390/genes13020332_

Round 1

Reviewer 1 Report

Dolcetti and colleagues reported on brain-derived neurotrophic factor (BDNF) Val66Met polymorphism in a group of 218 multiple sclerosis patients. Authors run principal component analysis showing association with the overall immune profile, but failed to find any actual clinical correlate. The manuscript is overall clear and well written. Laboratory methods are sound, while MRI analyses have been poorly explained. As such, while the topic is potentially interesting, on the current ground, authors only showed that BDNF Val66Met polymorphism is associated with immune profile of multiple sclerosis (as expected), in the absence of clinical correlates.

The population was enrolled between 2010 and 2019, and, thus, I assume a minimum follow-up of 2 years is available. Could you please include clinical and MRI follow-up to explore correlates of BDNF Val66Met polymorphism?

While there is a statistical difference between AA/AG and GG in disease duration, this is not clinically meaningful (two months is not big difference for a chronic disease, especially looking at disease duration distribution). I would suggest this difference is interpreted cautiously in both the abstract and the main body of the manuscript.

In the methods, authors describe a number of MRI sequences which potentially led to different MRI outcome measures. However, at the very end, they only reported on cortical thickness and lesion load. I wonder what additional measures are available and why they have not been included.

In the discussion, authors mention “the association between Val66Met and lower cortical volumes”. However, looking at results and figure 3, I would say that Val66Met patients have larger cortical thickness (and, thus, I also assume larger cortical volume, though this was not computed). Could you please double-check your results and revise the paper accordingly?

In the abstract, authors mention that “Analysis of MRI measures evidenced reduced cortical thickness in Met carriers at the time of diagnosis”. Could you please revise to Val66Met?

What is the added value of running this study on newly diagnosed patients? I see that “No corticosteroids or disease modifying therapies were administered before LP.” How long was that?

Author Response

Dolcetti and colleagues reported on brain-derived neurotrophic factor (BDNF) Val66Met polymorphism in a group of 218 multiple sclerosis patients. Authors run principal component analysis showing association with the overall immune profile, but failed to find any actual clinical correlate. The manuscript is overall clear and well written. Laboratory methods are sound, while MRI analyses have been poorly explained. As such, while the topic is potentially interesting, on the current ground, authors only showed that BDNF Val66Met polymorphism is associated with immune profile of multiple sclerosis (as expected), in the absence of clinical correlates.

A: Thank you for your kind summary and useful remarks. Here follows a point-by-point reply to the Reviewer’s comments.

Q1. The population was enrolled between 2010 and 2019, and, thus, I assume a minimum follow-up of 2 years is available. Could you please include clinical and MRI follow-up to explore correlates of BDNF Val66Met polymorphism?

A: Thank you The Reviewer's comment highlighted a typo in the Methods section. We apologize for the error; in fact the recruitment period is 2016-2019. We have corrected the sentence in the manuscript at line 98. The Reviewer's comment is absolutely appropriate. Unfortunately, due to lack of complete follow-up radiological and clinical data we could not investigate in the present study the impact of Val66Met on MS course. We agree that this issue is extremely important, accordingly, in the discussion is stated that “The lack of prospective MRI and clinical data is a major limitation of the present study and further research is needed to assess the impact of Val66Met polymorphism during the course of MS” (lines 375-377).

Q2. While there is a statistical difference between AA/AG and GG in disease duration, this is not clinically meaningful (two months is not big difference for a chronic disease, especially looking at disease duration distribution). I would suggest this difference is interpreted cautiously in both the abstract and the main body of the manuscript.

A: Thank you for this helpful comment that gives us the opportunity to improve the presentation and discussion of our findings. We agree that the slight difference in disease duration between the two groups does not properly represent a result of our study, but rather a possible (albeit minimal) confounding factor to be taken into account in the analyses. Therefore, we have removed this data from the abstract, and we have better explained in the Results that we have corrected the analyses considering this small difference (line 210-212; line 281-282).

Q3. In the methods, authors describe a number of MRI sequences which potentially led to different MRI outcome measures. However, at the very end, they only reported on cortical thickness and lesion load. I wonder what additional measures are available and why they have not been included.

A: Thank you for this comment. Although the 3T MRI protocol is standardized and potentially extendable to several parameters, in line with previous studies we have focused on cortical thickness as a measure of neurodegeneration in MS. Furthermore, the low number of patients with this additional MRI data did not allow us to investigate multiple parameters. Indeed, the study of other MRI measures, and in particular atrophy in specific brain regions may be particularly interesting. We have added a comment on this limitation in the discussion (line 370-375). Furthermore, we have improved the description of MRI methods as suggested by the Reviewer (lines 143-144,151-152).

Q4. In the discussion, authors mention “the association between Val66Met and lower cortical volumes”. However, looking at results and figure 3, I would say that Val66Met patients have larger cortical thickness (and, thus, I also assume larger cortical volume, though this was not computed). Could you please double-check your results and revise the paper accordingly?

A: Thank you for this comment. We apologize for this incomprehension. We have provided to modify the term "cortical volumes" in the manuscript with that of "cortical thickness", certainly more appropriate (line 368). As shown in Figure 3, patients carrying the Met had lower cortical thickness at diagnosis comparing with the Val/Val group (Met carriers, median [IQR]: = 2.6 [2.5-2.7] vs Val/Val, median [IQR] = 2.7 [2.6-2.8], p = 0.042). These data are in line with previous findings in healthy subjects (Egan et al., 2003).

Q5. In the abstract, authors mention that “Analysis of MRI measures evidenced reduced cortical thickness in Met carriers at the time of diagnosis”. Could you please revise to Val66Met?

A: Thank you for the observation. We modified the sentence according to the Reviewer’s suggestion (line 44).

Q6. What is the added value of running this study on newly diagnosed patients? I see that “No corticosteroids or disease modifying therapies were administered before LP.” How long was that?

A: Thank you for the reply. One of the aims of the present study was to explore the association between BDNF rs6265 polymorphism and central inflammation in MS, which has never been investigated previously. As CSF collection in MS is performed only for diagnostic purposed (OCB detection), all CSF samples were collected at the time of diagnosis. This is a general limitation of studies exploring CSF biomarkers in MS as repeated lumbar punctures are usually not performed for ethical reasons. We clarified this aspect in the discussion, at line 374-377. Concerning treatment with corticosteroids or DMTs, we have better specified in the Methods that no patient received such treatments before CSF collection.

Reviewer 2 Report

The BDNF Val66Met polymorphism modulates inflamma-2 tion and neurodegeneration in the early phases of multiple 3 sclerosis.

In this article Dolcetti Ettore et al.  investigated the role of polymorphic variant Val66Met of BDNF gene in early phases of multiple sclerosis.  The study  idea is general interesting, the number of the studied group is sufficient, but unfortunately I have a few of comments.

  1. In my opinion, the authors should use the correct notation for the Val66Met polymorphism in the BDNF gene in the body of the article (based on HGVS). The current transcript is only a record of the effect of single nucleotide polymorphism at the protein level. If the authors present genotyping results, all necessary information describing the variant should be included in the text.

  1. Material and methods:

- the information placed in this section does not indicate that there were only women in the MS patient group. If so, why don't the authors write about it ? Why were there only women in the group analyzed ? Readers might think that the disease only affects women, but we know that this is not the case. Actually it affects men less often but if the authors collected the group from 2010-2019 were there no men with MS at that time ?

- what race were the patients in the study group from ? This is important because of population differences in the occurrence of SNP-type polymorphisms;

- i don't quite understand which method of detecting the polymorphism under study was used by the authors ? First they write about using TaqMan molecular probes, which is commonly used (the choice of device is adequate) and then they write about using Sanger sequencing ? If the authors used a Sanger-type method then why don't they provide the reference sequence number of the BDNF gene they used to analyze the results ? What tool did the authors use to design the primers ?

  1. Results:

- what purpose did the authors have in presenting genotyping results in the form of creating subgroups with AA and AG genotypes ? It is unreadable and does not give the full picture (this relates to tables and logistic regression results of the association of polymorphisms and cytokine levels). Also, indicate which genotype is the reference. Also missing, in my opinion, is the provision of accurate information on the frequency of both alleles in this group;

- I disagree with the authors that we can describe genetic test results in terms of „Met alleles” and „Val alleles”. Sorry, but from a genetics point of view and in this type of Journal this should not happen !

Author Response

The BDNF Val66Met polymorphism modulates inflammation and neurodegeneration in the early phases of multiple sclerosis.

In this article Dolcetti Ettore et al.  investigated the role of polymorphic variant Val66Met of BDNF gene in early phases of multiple sclerosis.  The study idea is general interesting, the number of the studied group is sufficient, but unfortunately I have a few of comments.

A: We thank the Reviewer for his important remarks. We tried to improve the manuscript following the Reviewer’s suggestions. Here follows a point-by-point reply.

Q1. In my opinion, the authors should use the correct notation for the Val66Met polymorphism in the BDNF gene in the body of the article (based on HGVS). The current transcript is only a record of the effect of single nucleotide polymorphism at the protein level. If the authors present genotyping results, all necessary information describing the variant should be included in the text.

A: Thank you for this comment that will certainly improve the quality of work. We modified the specific section, adding more detail about this polymorphism in the Introduction (line 74-80). The correct annotation is: NM_001143810.1:c.[ 196G>A];NP_001137282.1 p.(Val66Met) (rs6265) on ref genomes (GRCh38/hg38) (Human Genome Variant Society,http://www.hgvs.org/mutnomen)

Q2. Material and methods:- the information placed in this section does not indicate that there were only women in the MS patient group. If so, why don't the authors write about it? Why were there only women in the group analyzed? Readers might think that the disease only affects women, but we know that this is not the case. Actually it affects men less often but if the authors collected the group from 2010-2019 were there no men with MS at that time?

A: We apologize for the typo in the Methods section, we provided to correct the enrollment period which is “between 2016 and 2019” and not 2010-2019 as previously stated. Regarding the sex distribution in our cohort of patients, our sample population fully reflects the characteristics of a typical MS patients population at diagnosis, consisting mostly of women (in our sample 155 out of 218 total individuals, i.e., 71.1%) and to a lesser extent men (63 out of 218 individuals, i.e., 28.9%) who were included in the analysis. For completeness we have further explained this aspect in the respective tables (line 190-191; 203-204)

Q3. what race were the patients in the study group from? This is important because of population differences in the occurrence of SNP-type polymorphisms;

A: Thank you for your kind reply. We have better specified that patients enrolled came from Central and Southern Italy and we compared allele frequencies with data of European population. We specified this aspect in Methods (line 96) and in Results (line 194-196) 

Q4. I don't quite understand which method of detecting the polymorphism under study was used by the authors? First they write about using TaqMan molecular probes, which is commonly used (the choice of device is adequate) and then they write about using Sanger sequencing? If the authors used a Sanger-type method then why don't they provide the reference sequence number of the BDNF gene they used to analyze the results? What tool did the authors use to design the primers?

A: Thank you for kind answer. We added in Methods the requested details on TaqMan Genotyping Assay (line 110-117)

Q5. Results: - what purpose did the authors have in presenting genotyping results in the form of creating subgroups with AA and AG genotypes? It is unreadable and does not give the full picture (this relates to tables and logistic regression results of the association of polymorphisms and cytokine levels). Also, indicate which genotype is the reference. Also missing, in my opinion, is the provision of accurate information on the frequency of both alleles in this group;

A: Thank you for reply. The need to combine the different genotypes into subgroups is due to the fact that the AA group, as happens in the European reference population, is poorly represented in our sample and therefore cannot be individually analyzed. Moreover, the presence of a single allele coding for Met in this polymorphism produces alterations in the function of the neutrotrophin itself (Egan et al. 2003). Furthermore, in many works carried out on the European population, the Val66Met different genotypes have been combined into subgroups in order to be effectively analyzed (Portaccio et al., 2021; deMeo et al., 2021; Fera et al.,, 2013). We provide to name the two groups respectively Met carriers (risk allele) and Val/Val (non-risk allele). We explained the frequency of both alleles in the two groups (line 198-200)

Q6. I disagree with the authors that we can describe genetic test results in terms of „Met alleles” and „Val alleles”. Sorry, but from a genetics point of view and in this type of Journal this should not happen !

A: Thank you for reply. We provided to correct in manuscript

Round 2

Reviewer 1 Report

Authors have addressed my concerns